# An Assessment of the Social Costs of Road Traffic Crashes in Cameroon

Peter Taniform [1], Luca Persia [2], Davide Shingo Usami [2,*], Noella Bajia Kunsoan [2], Mary M. Karumba [3] and Wim Wijnen [4]

1   World Bank Group, Nairobi P.O. Box 30577-00100, Kenya
2   Centre for Transport and Logistics, "Sapienza" University of Rome, 00184 Roma, Italy
3   State Department for Planning and Statistics, Nairobi P.O. Box 30005-00100, Kenya
4   W2Economics, 3523 Utrecht, The Netherlands
*   Correspondence: davideshingo.usami@uniroma1.it; Tel.: +39-06-44585134

**Abstract:** This study estimated the social costs of road traffic crashes (RTCs) in Cameroon, motivated by a lack of empirical evidence for economic loss and social suffering associated with RTCs menace in developing countries particularly Sub-Sahara Africa. A model for estimation of cost based on a combination of valuation methods was developed following international guidelines, and can be adapted for other developing countries similar to Cameroon's context. Five cost components were estimated namely: production loss; human costs; medical costs; property damage costs and administrative costs. Data from the field, secondary databases and transfer values were used together with adjustments for under-reporting of road traffic crash data that is prevalent particularly in developing countries. Total social cost of RTCs in Cameroon in 2018 was USD 3.6 Billion and is equivalent to 3.8% of GDP in 2018. This estimate is way above RTCs cost estimates obtained by studies in Sub-Sahara Africa using the human capital approach, and slightly outside the range of social cost estimates found in Lower- and Middle-Income Countries (LMICs) literature. The estimate is also larger than the conservative figures used for policy purposes such as the current National Road Safety Strategy, implying that under-reporting of RTCs data under-represents apparent socio-economic value of RTCs. The study recommends improvement in the procedures of crash data by operationalizing the recently established centralized RTCs database, as well as adoption of systematic approaches to estimation of crash costs by policy makers.

**Keywords:** social costs; road traffic crash; human capital approach; willingness to pay

## 1. Introduction

Road Traffic Crashes (RTCs) are the 8th cause of human death, with 90% of the deaths resulting from road traffic injuries (RTIs) occurring in Lower- and Middle-Income Countries (LMICs) [1]. Cameroon is one of the LMICs with the highest RTC fatality rate, with 30.1 persons per 100,000 population in 2016 [2], higher than the average for both low- and middle-income countries. The largest proportion of Cameroon's population also falls within the age bracket (5–29 years), whose leading cause of death is RTCs. Improving road safety may contribute to future gains in health of this population. Road safety issues in Cameroon are further aggravated by insufficient policy measures [3], with lack of reliable social cost estimates of RTCs being one of the multifacet factors that hinders proper road safety policy making. According to Wijnen & Stipdonk [4] information about the social cost of RTCs is an important input to formulation of road safety policymaking. This is because the social costs are indicative of some of the repercussions on the economy and its people of road traffic deaths, injuries and property damage. The quantified costs may therefore inform trade-off in the allocation of national resources to competing needs and interests, potentially leading to effective decision-making in the allocation of resource to road safety strategies.

Besides providing reliable and transferable estimates of social costs of RTCs in Cameroon, this study attempts to close the gaps observed in previous studies carried out in similar contexts of Cameroon, especially for Sub- Saharan Africa (SSA). In line with international best practices, the study proposes a combination of methods i.e., the human capital, willingness to pay, and restitution costs to estimate the social costs of RTCs in Cameroon.

## 2. Literature Review

Despite the huge and unbalanced burden of RTCs in developing countries and Cameroon, Prakash et al. [5] and Bougna et al. [6], among others note that there is little to no empirical evaluation of the implication of RTCs in developing regions of the world such as Sub-Sahara Africa (SSA). This is occasioned by a lack of exhaustive and consistent databases on RTCs and other information required for assessing the social cost of RTCs [7]. Lack of systematic evidence on the social cost of RTCs leads to ad hoc and often inadequate allocation of resources for road safety, as illustrated by the revised National Road Safety Strategy in Cameroon. The latter has increased commitments to road safety including the Sustainable Development Goal (SDG) 3.6, but funding of the same has been retained at 4% of the National Road Fund Levy. This tendency of allocating resources for road safety without evidence fails to follow recommended practices of road safety management [8] and may sustain the current trends in RTC. Assuming the pattern predicted for RTCs in Sub-Sahara Africa countries by World Bank [9], Cameroon's RTCs fatality rate could double by 2030 following a business-as-usual approach to the management of RTCs. To contribute to the policy debate on resource requirements for reducing RTCs in Cameroon, there is a need to quantify the socio-economic losses attributable to RTCs. This study makes the first attempt to develop a framework and estimate the social costs of RTCs in Cameroon, which does not exist to the best of our knowledge.

Globally, the total cost of RTCs has been estimated to range between 1.1% to 2.9% of Gross Domestic Product (GDP) for LMICs and 1.7% to 6% of GDP for Higher Income Countries (HIC) [4]. According to Bougna et al. [6], two main methods are employed for quantifying RTCs costs in literature: Willingness to Pay (WTP) and Human Capital (HC). In addition, the restitution costs method is commonly used to quantify several cost components [10,11]. The WTP method elicits the amount of money individuals are prepared to pay for reducing the chances of dying in an RTC, thereby estimating the value of statistical life (VOSL). This is done by either asking respondents in a survey, directly or indirectly, how much money they are willing to pay for reducing fatal crash risk (stated preferences). This amount can be found, for example, by asking respondents to choose from different routes concerning risk and travel costs [12]. Also, it can be observed how much they have already invested in measures aimed at reducing the chance of dying in an RTC (revealed preferences). The HC method quantifies the resources (time and money) lost or foregone upon occurrence of an RTC. The method is more popular in developing country literature, because it uses straightforward inputs to calculate cost estimates that are easy to translate for policy purposes, [4,6,13]. On the other hand, the WTP is difficult to use in the LMICs context due to doubts over population's ability to trade-off risk and wealth [4]. The restitution costs method is aimed at calculating the costs that are made to restore road casualties and their relatives and friends as much as possible to the situation which would exist if they had not been involved in a road crash [10]. This method is suitable for estimating medical costs and property damage, among others.

Few studies have recently attempted to estimate the cost of RTCs in specific contexts within Sub- Saharan Africa (SSA). Prakash et al. [5]; Mofandal & Kaniptong [13]; Abdalla, Hakim, Wahdan & El Refaye [14]; Labuschagne, De Beer, Roux & Venter [15] and Parkinson, Kent, Aldous, Oosthuizen & Clarke [16] and Murad [17] employ different approaches to estimate the cost of RTCs in Mozambique, Sudan, Egypt, South Africa, Mozambique and Ethiopia, respectively. These studies reflect cost quantities with a wide variance that is partly attributable to the data available, estimated cost components and estimation approaches making the estimated social cost of RTCs difficult to use for the case of Cameroon. Prakash

et al. [5] and Parkinson, Kent, Aldous, Oosthuizen & Clarke [16] restrict themselves to medical costs of RTCs which is only one out the six components according to international best practice ([10,18]).

Another shortcoming in literature is that studies on RTCs social cost estimates, especially within Sub-Saharan Africa (SSA), do not adjust the RTCs data for under-reporting (see [13,15]). Under-reporting is a major concern that leads to under-estimation of RTCs social costs in LMICs contexts like Cameroon [2,19,20]. It is also apparent that most estimates of social costs of RTCs in LMICs fail to consider the costs associated with loss of quality of life, pain and anguish of RTCs. Where an attempt is made to estimate the same, the use of insurance compensation as proxies in under-developed insurance markets (see Mofandal & Kaniptong [13] for instance), leads to the under-estimation of social costs of RTCs. With these gaps observed in studies for SSA, ranging from wide variances in results, limitation of cost components evaluated, issues of under-reporting, and limitation to methodologies, there is a need to provide new and more so reliable costs estimates for the case of Cameroon.

This study adopts a combination of methods (human capital, willingness to pay and restitution costs) in estimating the social costs of RTCs, following international guidelines and best practices ([10,11,18,21]). This is a first attempt for Cameroon, to the best of our knowledge. In the current study, we correct for under-reporting of crashes, fatalities and injuries from road crashes to side-step shortcoming of previous empirical work that attempts to estimate the social cost of RTCs. Further, a recently developed value transfer method is used to calculate human costs based on Milligan, Kopp, Dahdah, & Montufar [22]. This function estimates the value of a statistical life based on willingness to pay, using GDP per capita. The model developed within the framework of this study is applicable within other African countries of similar context as Cameroon.

## 3. Methodology

This section describes the cost elements considered for this study, the formulas used to estimate them and the data that was used.

### 3.1. Cost Elements

The socio-economic costs considered for this study were those identified in the international literature such as Wijnen et al. [10]; World Bank [23]; Trawén, Maraste, & Persson [24]; Alfaro et al. [11] for which data was available for Cameroon. The costs are classified into six major cost components following Wijnen et al. [25] and Kasnatscheew et al. [26]. Furthermore, a distinction is made between injury-related costs and crash-related costs, as summarized in Figure 1.

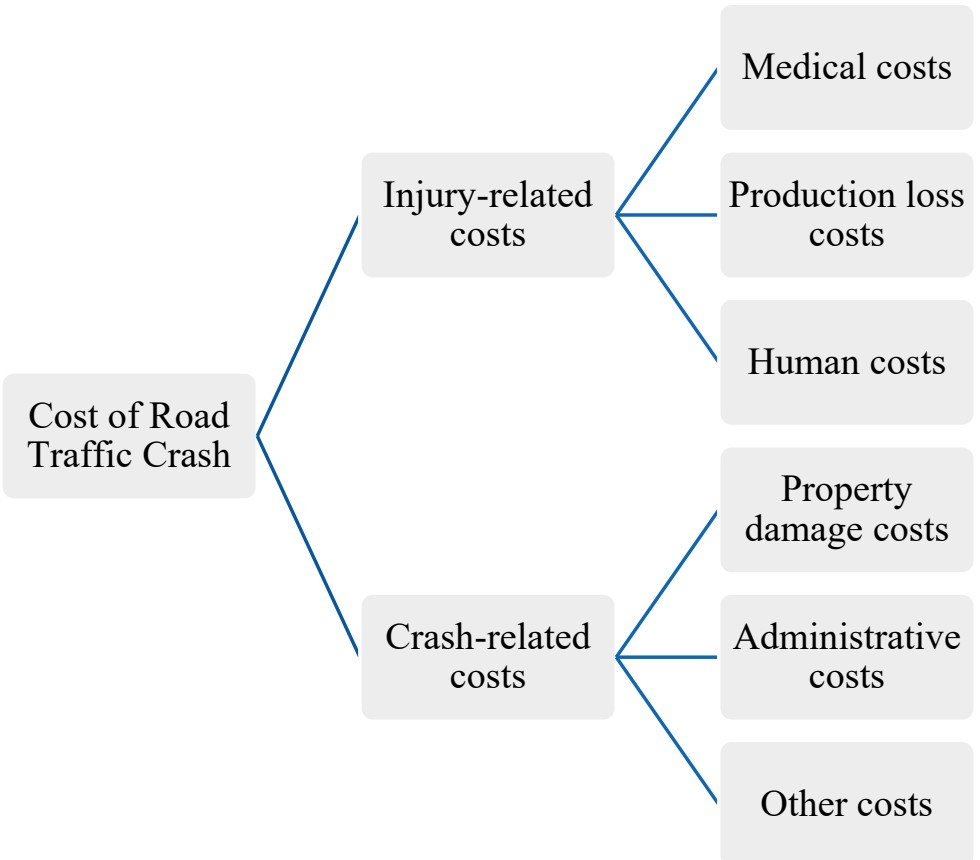

**Figure 1.** Classification of the socio-economic costs of RTCs. Source: Adopted from Wijnen et al., (2017) [25].

### 3.2. Cost Element and Estimation Method Used

Human costs of RTCs were estimated using WTP approach to calculate the Value of Statistical Life (VoSL). Surveys to elicit WTP were not possible for the current study, and instead, a benefit transfer function for LMICs developed by Milligan et al. [22] was adapted.

Production costs were estimated using the Human Capital method. The method establishes the losses occasioned by death or injury of humans, based on withdrawal of the labour hours and wage they expended while alive or healthy.

The other cost elements (property damage, medical and administrative costs) were estimated using restitution method. This incorporates the tangible costs such as services, expertise, materials, tools & equipment involved in treating victims, repairing vehicles and pursuing administrative procedures to return the victims to normal life after accident (insurance claim process, court process, among others)

Below we present the components of cost elements and the formulas used to calculate the costs (adapted from [27]).

1.  **Medical costs:** these are ambulance ride costs; in-patient treatment and hospital stay costs, and out-patient medical costs. The formulas used in the model for estimating each of these sub costs are shown in Equations (2)–(4), and are used in calculating the total medical cost in Equation (1).

$$\mathbf{TMC = AMBC + HCIN_{sev} + HCOUT} \tag{1}$$

where:

> *TMC = Total Medical Costs*
> *AMBC = Ambulance costs*
> *HCIN$_{sev}$ = hospital in-patient and ward stay costs*

*HCOUT = out-patient treatment costs*

$$\mathbf{AMBC = PATRC * NCC * ACAMB} \tag{2}$$

where:

*AMBC = Ambulance costs*
*PATRC = Proportion of casualties arriving by ambulance*
*NCC = Number of casualties arriving in the accident emergency department*
*ACAMB = Average costs per ambulance trip per casualty*

$$\mathbf{HCINsev = CASsev * HDURsev * DHCIN} \tag{3}$$

where:

$HCIN_{sev}$ = *hospital in-patient treatment and ward stay costs*
$CAS_{sev}$ = *number of hospitalized casualties by severity (fatality (F) or serious injuries (SI))*
$HDUR_{sev}$ = *average duration of hospitalization by severity*
*DHCIN = daily costs in-patient treatment per casualty*

$$\mathbf{HCOUT = OUTINJ * ACOUT} \tag{4}$$

where:

*HCOUT = hospital costs out-patient treatment*
*OUTINJ = number of out-patient casualties (treated at emergency department)*
*ACOUT = average costs of out-patient treatment per casualty*

2. **Production (gross) loss** resulting from lost work hours due to death and injuries of victims, estimated according to Equation (5). This is assumed to incorporate a consumption loss, since the wages people are paid from working are applied for consumption purposes. It is common to adjust the production loss for costs due to pain, suffering, grieve and inconvenience, but due to data unavailability, these costs were not factored in the equation. This is likely to under-estimate the total costs.

$$\begin{aligned}[\mathbf{PLF_{age,\,gnd} + PLPD_{age,\,gnd} + PLINJ}] \\ = \textstyle\sum_{\mathbf{age,\,gnd}}(\mathbf{PD_{age,\,gnd} * PPL_{age,\,gnd}}) + \sum_{\mathbf{age,\,gnd}}(\mathbf{F_{age,\,gnd}} \\ \mathbf{*PPL_{age,\,gnd}}) + \textstyle\sum_{\mathbf{sev,\,gnd}}\ (\mathbf{INJ_{sev,gnd} * DWAGE * WDUR_{sev,}})\end{aligned} \tag{5}$$

where:

$PLF_{age,gnd}$ = *total gross production loss from fatalities*
$PLPD_{age,gn}$ = *total production loss permanently disabled*
*PLINJ = production loss from injuries (temporarily unable to work)*
$PD_{age,gnd}$ = *number of permanently disabled*
$F_{age,gnd}$ = *number of fatalities*
$PPL_{age,gnd}$ = *Per person Production loss*
$INJ_{sev,gnd}$ = *number of injuries by severity (serious, slight)*
*DWAGE = gross wage and other employee related costs/day*
$WDUR_{sev}$ = *duration of absence from work by severity (number of days)*

3. **Human Costs,** which include the intangible costs of lost life years and reduced quality of life from pain and grief resulting from RTCs by victims or relatives, were estimated as a sum-total of the cost from fatalities, serious injuries and slight injuries (Equation (6)). To estimate the human cost of fatalities, a value transfer function for LMIC as presented by Milligan et al. [22], was used to estimate the value of a statistical life (VoSL) in Cameroon. Consumption loss resulting from fatalities is subtracted from the VoSL, since this component is included in the gross production loss function (see Equation (7)). Borrowing from Wijnen [28], human costs of slight and serious injuries were estimated as a percentage (13% and 1%) of VoSL, while assuming negligible (zero) consumption loss from injuries (see Equations (9) and (10))

$$HC = HC_f + HC_{SI} + HC_{slightI} \tag{6}$$

$$HC_f = F * (VoSL - CL)$$
$$VoSL = \left\{ \left[ 1.3732e^{-4} * GDP_{p.c.}^{2.478} \right] \right\} \tag{7}$$

$$CL = \sum_{age=0}^{age=99} PCL_{age\_gnd}$$
where
$$PCL_{age,\,gnd} = \sum_{t=0}^{T} C * \frac{1}{(1+r)^t} \tag{8}$$
and $\quad T = LE_{age,gnd} - AGE$

$$HC_{SI} = 0.13 * SI * VoSL \tag{9}$$

$$HC_{SLI} = 0.01 * SLI * VosL \tag{10}$$

where:
  $HC_f$ = Human costs (fatalities)
  $F$ = Number of casualties (fatalities)
  $SI$ = Number of casualties (serious injuries)
  $SLI$ = Number of casualties (Slight Injuries)
  $VoSL$ = Value of a statistical life
  $GDP_{p.c}$ = Gross Domestic Product per capita
  $CL$ = Average consumption loss per person over remaining years
  $PCL_{age\_gnd}$ = discounted consumption loss per person, by age and gender
  $C$ = per capita private consumption
  $T$ = remaining life years of a person (to life expectancy)
  $HC_{SI}$ = Human costs (serious Injuries)
  $HC_{SLI}$ = Human costs (slight Injuries)

4.  **Property damage cost**: the damage to motor vehicles is estimating using Equation (11). The cost of other property and road damages is not included due to limited data.

$$PDM = \sum_{sev,\,type} \left( AMVD_{sev,type} * MV_{sev,type} \right) \tag{11}$$

where
  $PDM$ = motor vehicle damage costs
  $AMVD_{sev,type}$ = average motor vehicle damage costs by vehicle type and crash severity
  $MV_{sev,type}$ = the number of motor vehicle damaged by type and crash severity

5.  **Administrative costs:** The costs comprise of police (Equation (12)), fire service (Equation (13)), insurance (Equation (14) and legal/judicial costs (see Equation (15). These costs add up to give the total administrative costs.

$$PC = \sum_{Sev} (CR_{sev} * PA_{sev} * PTIME_{sev} * PWAGE) \tag{12}$$

where:
  $PC$ = Police costs
  $CR_{sev}$ = Number of crashes by crash severity
  $PA_{sev}$ = Percentage of police attendance by crash severity
  $PTIME_{sev}$ = Police time spending (hours) by crash severity
  $PWAGE$ = Average wage of a police officer per hour

$$FC = \sum_{Sev} (CR_{sev} * FA_{sev} * FTIME_{sev} * FWAGE) \tag{13}$$

where:
  $FC$ = Fire service costs
  $CR_{sev}$ = Number of crashes by crash severity

*FA$_{sev}$ = Percentage of fire service attendance by crash severity*
*FTIME$_{sev}$ = Fire officer time spent (hours) by crash severity*
*FWAGE = Average wage of a fire officer per hour*

$$\mathbf{IC} = \sum\nolimits_{\mathbf{Sev}}(\mathbf{IAC/C_{Sev}} * \mathbf{Prop} * \mathbf{CR_{Sev}}) \tag{14}$$

where:

*IC = Insurance admin costs*
*IAC/C$_{Sev}$ = The average insurance administrative costs by crash severity*
*Prop = Proportion of motor vehicle insurance policies claims declared*
*CR$_{sev}$ = Number of crashes by crash severity*

$$\mathbf{JCAC} = \sum\nolimits_{\mathbf{Sev}} \mathbf{CC/C_{Sev}} * \mathbf{CR}_{Sev} \tag{15}$$

where:

*JCAC = Judicial Administrative costs*
*CC/C$_{Se}$ = The average cost of administering traffic-related court case*
*CR$_{sev}$ = Number of crashes by crash severity*

## 4. Data

This section explains the procedures used to correct road safety data for under-reporting before using it in the model for estimating the total costs of RTCs and the data used for the assessment of the socio-economic costs of RTCs.

### 4.1. Road Crash Causalities Data

There are three official and independent sources of statistics on the number of road casualties in Cameroon: the National Gendarmerie (a police force with a military order), the National Police and the Ministry of Public Health. The Ministry of Health hospital-based data covers cases of crash fatalities or injuries that end up in hospitals across the country, with its availability stretching back to the year 2018. On the other hand, the Gendarmerie and Police data cover rural and urban jurisdictions, respectively, making these two databases complementary. This also implies that some RTC cases may end up in one source and not the other, due to a lack of centralized command. The hospital data are likely to overlap partly with the Gendarmerie and Police data, as part of the injured people treated in the hospital are involved in crashes reported to the Gendarmerie or Police. However, other injuries may occur in crashes which are not reported to the Gendarmerie or Police but are treated in hospital and therefore included in hospital data by the Ministry of Public Health. The year 2018 was chosen as the reference year for the calculation of social costs, meaning that all data and results refer to this year. The number of RTC fatalities and injuries is recorded in Table 1.

**Table 1.** Number of casualties as recorded by the Police, Gendarmerie and Ministry of Health (2018).

|  | Police | Gendarmerie | Total | Ministry Health |
|---|---|---|---|---|
| Fatalities | 583 | 782 | 1365 | 1286 |
| Injuries | 1759 | 2801 | 4560 | 126,306 |

Source: Authors compilation from Agency data (Police, Gendarmerie and Ministry of Health).

A striking observation in Table 1 is the much higher number of injuries as recorded by hospitals in comparison to Police and Gendarmerie data, also observed in recent studies like Niditanchou, Plamar & Janz [19]. This is likely to be explained by Police and Gendarmerie failure to report all injuries or casualties not reporting injuries. For purposes of this study, hospital data is much better at estimating non-fatal injuries than police data. We acknowledge that some injuries caused by RTCs are not treated in hospital, but rather by traditional healers or even basic home care. There is little information about these kinds of

injuries and their related costs, and for purposes of this study, we assume that the impact of these slight injuries on the total costs is relatively low and therefore they will not be included in this study.

RTCs injury statistics from the Ministry of Public health are not disaggregated according to the various levels of injury severity. A pattern observed in McGreevy [29] in a hospital-based survey of RTCs injury victims was used to separate the number of injuries into two main levels of severity: Slight (73%) and Serious (27%) injuries as shown in Table 2. The Ministry of Public Health data in Table 1 is disaggregated further as shown in Table 2.

*4.2. Adjusting RTCs Hospital Data for Under-Reporting*

As highlighted by the literature, the number of RTC casualties in Cameroon is likely to be underestimated. The level of under-reporting will differ depending on the data source considered (Hospitals, Police, Gendarmerie) and the level of severity of the injuries [30]. The under-reporting rate is ideally estimated using a capture-recapture method [7] linking two different databases. This is a complex method which requires the availability in the two databases of sufficient data to make the link, which is not the case for Cameroon. As an alternative, in this study the WHO estimates [1] are used to estimate the under-reporting rate of fatalities. WHO uses a negative binomial regression model to estimate road fatalities in countries without eligible death registration data like Cameroon. In 2016, the number of fatalities reported by the Ministry of Transport in Cameroon [31] to the WHO was 1879. Using WHO regression-based methods and variables that are theoretically relevant as predictors of fatality rates, an estimate of 7066 fatalities was made with a 95% confidence interval of (5670–8463). This implies that the reported fatalities in 2016 were between 22% and 33% of the estimated ones, with a best estimate of 26.5%. Similar values were found in another report [32], with a reporting rate for Cameroon in 2010 between 20% and 25%. Other studies in African countries using the capture-recapture method suggest reporting rates of police records ranging from 31% to 68% [30]. Values vary from country to country based on existing conditions. Since other data sources useful to understand the fatality reporting rate are not available in Cameroon, the value of 26.5% has been considered in this study. It is also assumed that the under-reporting rate has not changed two years later, since there are no significant policy changes implemented relating to data collection. Therefore, the reported total fatality (1365) data covers only 26.5% of the fatalities in 2018. This implies that the number of fatalities adjusted for under-reporting in 2018 was 5151.

The WHO does not address under-reporting rates for injuries, and this study imputed the same from the under-reporting rate of fatalities. A large-scale hospital survey by Kourouma et al. [33] in Guinea suggests that approximately half of fatalities occur in hospitals and are therefore captured in hospital records. This country belongs to the same socio-economic category as Cameroon, and the observed tendency is likely to be close to what would be observed in Cameroon. Based on this, the reporting rate for injuries was put at double (26.5 × 2 = 53%) that of fatalities. This value is slightly lower than hospital registry-based reporting rates estimated for two other African countries, Ethiopia (55%) and Uganda (60%) [30]. Adoption of higher reporting rates for injuries would effectively lead to lower estimated cost of RTCs. The under-reporting rates of RTC casualties by severity are summarized in Table 2. It is assumed in this study that slight injuries describe RTC injury victims who were treated and discharged in the same day, largely based on expert explanations of the data. Furthermore, out of the seriously injured victims, Mofandal & Kaniptong [13] indicate that 17.5% ended up with permanent disabilities (see Table 2).

**Table 2.** Number of road casualties in 2018 by the level of severity—before correcting for under-reporting.

| Level of Severity | Number of Victims | Under-Reporting Rate | Estimated Number of Victims (Adjusted for Under-Reporting) |
|---|---|---|---|
| Fatalities | 1365 | 73.5% | 5151 |
| Serious injuries—Non-permanent injury | 28,135 | 47% | 53,085 |
| Serious injuries—permanently injury | 5968 | 47% | 11,260 |
| Slight Injuries | 92,203 | 47% | 173,968 |

Source: Authors compilation from Agency data (Police, Gendarmerie and Ministry of Health).

### 4.3. Number of Crashes

Estimates of the number of crashes are needed for cost calculations, as several cost elements are calculated on the basis of the number of crashes [4]. There were two sources of data on road traffic crashes (the Gendarmerie and Police databases), and there is no existing documentation for the rate of crash under-reporting. For this study, the number of fatal and injury crashes was estimated based on casualty-to-crash ratios calculated from Police Department's data for 2016 (see Table 3). The police data quality for 2016 was judged to be better than for other recent years due to expert opinion and the pattern in panel A of Figure A1 in Appendix A.

**Table 3.** Casualty to Crash ratio using police data for fatal and injury crashes in 2016.

| | Number of Casualties | Number of Crashes | Ratio |
|---|---|---|---|
| Fatal | 721 | 607 | 1.19 |
| Injury | 2886 | 2175 | 1.33 |

Source: Authors calculations from Agency data (Police) data.

The ratios (as computed in Table 3) were then applied to the 2018 Ministry of Public Health data after correcting the same for under-reporting.

As far as under-reporting of PDO crashes is concerned, the study used Police department data (see Table 4) to estimate the total ratio of injury casualties to PDO crashes (6066/2886 = 2.1) that was applied to the adjusted number of serious and slight injuries by Ministry of Public Health for 2018 (238,313). From Table 4, the number of estimated PDO crashes in 2018 is half a million (500,457).

**Table 4.** Number of crashes in 2016 as reported by the Police Department, Gendarmerie and Total Crashes before and after adjusting for under-reporting.

| Level of Crash Severity | Police | Gendarmerie | Total Crashes | Ratio of Casualties to Crashes | Adjusted Number of RTCs Casualties Used (See Table 2) | Number of Crashes (2018) after Adjusting for Underreporting |
|---|---|---|---|---|---|---|
| Fatal Crash | 607 | 896 | 1503 | 1.19 | 5151 | 4337 |
| Injury Crash (serious and slight) | 2175 | 1230 | 3405 | 1.33 | 238,313 | 179,602 |
| Property Damage Only (PDO) crashes | 6066 | 896 | 6962 | 2.1 * | 238,313 | 500,904 |

Source: Agency data (Police and Gendarmerie) from Cameroon and authors' estimations (* Ratio PDO crashes to the total number of injuries).

The ratios in Table 4 (number of casualties to the number of crashes) are in line with the ratios used in road crash cost studies in other African countries, as shown in Table 5, which provides some support for the reliability of our estimates.

**Table 5.** Ratio of number of casualties to number of crashes in African road crash cost studies.

| Study | Country | Ratio of Number of Casualties/Number of Crashes | | |
|---|---|---|---|---|
| | | Fatal | Serious Injury | Slight Injury |
| Labuschagne et al., 2017 | South-Africa | 1.29–1.33 | 1.37–1.43 | 1.23–1.38 |
| Modafal & Kanitpong, 2016 | Sudan | 1.18 | 1.84 | 1.94 |
| Murad, 2011 [17] | Ethiopia | 1.26 | 1.38 | 1.38 |
| Our study | Cameroon | 1.19 | 1.33 | 1.33 |

Source: Authors compilation from studies.

*4.4. Medical Costs*

To estimate medical costs, data was needed on the number and cost of casualties receiving specific types of medical services. Primary data from hospitals was not available for the purposes of this study. To overcome this, Hospital surveys from studies carried out in Cameroon [29]; Guinea [33] and Sudan [13] were used to establish the share of causalities by severity, unit cost and length of medical services.

*4.5. Production Loss*

The human capital approach used disaggregated demographic and labour market data obtained from the National Institute of Statistics and World Bank's WDI database. In addition, information on the period of time that injured victims are not able to work was obtained from Murad [17], a study that conducted household and casualty surveys in Ethiopia.

*4.6. Human and Consumption Costs*

GDP per capita in Cameroon was used to estimate the VoSL using the value transfer function following the steps in Milligan et al. [22]. First the GDP per capita in 2018 in local currency units was expressed in 2005 dollar to obtain the VoSL. The resultant VoSL was then converted to the 2018 local currency unit. On both occasions, the purchasing power parity (PPP) and GDP deflator index based on the World Development Indicators [34].

For purposes of calculating consumption loss from a fatality, the profile of remaining life years of a victim (had the fatality not occurred) was used based on demographic information (gender-based age structure and life expectancy) and discount rate. This was applied on the private consumption per capita, also derived from World Bank [34] and adjusted using metrics (PPP) from the same source. It was assumed that consumption loss from serious and slight injuries was negligible following Wijnen [28], and this is likely to under-estimate the costs.

*4.7. Vehicle Damage Costs*

Since it was not feasible to get a representative sample for the survey undertaken by this study, the average motor vehicle damage costs were obtained from Murad [17], while those of motorcycle damage costs were obtained from Kamzi & Zubair [35]. The process of converting the costs involved updating the costs (in local currency units) to the year 2018 from the respective data collected in these studies. The changes in Consumer Price Indices (CPI) from the World Bank [34] tables were used, and the prices were then converted to the international dollar and the XAF (Local currency in Cameroon), using the PPP from the same source. The resulting costs were validated by experts in insurance companies (loss valuers and assessors working in the insurance industry).

*4.8. Administrative (Police, Emergency Service, Legal and Insurance Costs)*

These are minor costs, largely accounting for the time that staff (judicial and insurance, fire service and police officers) spend attending to RTC-related matters. Police costs were obtained by conducting interviews with representatives of the National Gendarmerie and Police based in different areas (rural and urban). These are key informant interviews only.

Household surveys deployed for the study did not yield reliable data due to high non-response rate occasioned by political tensions in Cameroon. A similar approach at the fire service department was followed to obtain information on emergency services. The total police and fire service costs were calculated based on the average number of staff deployed to a site, time spent, average gross hourly wage across common cadre and the proportion of accidents that are attended by police. This study opted to use averages for purposes of estimation even if there may be issues of departure from normal distribution for two reasons. First an average is a representative number as it covers all points of observations. Second, other national-level data used are in the form of averages. So, for purposes of comparison of the study outcome with other current and future studies the average is a safer option than any other measure of central tendency.

Data on insurance administration costs were obtained from transferred values from a survey conducted in Sudan by Mofandal & Kaniptong [13]. The original costs were updated to 2018 prices using the CPI and converted to local currency units (XAF) using PPP exchange rates for the respective years. This data was moderated by weighting it using the proportion of motor vehicle insurance policies that are declared in Cameroon, as reported in both Fondzenyuy [36] and Association Des Societes D'assurances du Cameroun [37]. The same process was followed for judicial costs, where the average cost of processing a crash was estimated using information on resources spent gathered from key informant responses.

Details regarding other fine data required and the source of the same are described in Table A1 of the Appendix A. The characteristics of the countries from where transfer values were adapted are also presented in Table A2 of the Appendix A.

## 5. Results and Discussion

The total social costs of RTCs in 2018 for Cameroon, estimated from our model, is XAF 808.9 billion, equivalent to USD 3.6 Billion using PPP (XAF 225 per USD) for the year 2018. The model is based on the formulas and data laid out in Excel. Put into perspective, these costs are equivalent to 3.8% of the GDP of Cameroon in 2018, which was recorded as XAF 21,500 billion. The distribution of this cost based on the classification in Figure 1 is provided in the following sub-sections.

### 5.1. RTCs Injury-Related Costs

According to Table 6, the total injury-related cost of RTCs was estimated at XAF 346,528 million (cost in USD is in parenthesis throughout this section). The largest (38%) component of this cost is attributed to human costs, followed closely by production and medical costs.

**Table 6.** Estimated injury-related costs of RTCs in Millions of XAF and USD (Production, human and medical) per severity level.

|  | Medical Costs | Production Loss | Human Cost | Total Injury-Related Costs |
|---|---|---|---|---|
| Fatality | 182 (1) | 32,095 (143) | 45,040(200) | |
| Serious Injury | 100,391 (446) | 70,241 (312) | 73,142 (325) | - |
| Slight Injury | 7414 (33) | 2811 (12) | 15,212 (68) | |
| Total | 107,987 (479) | 105,147 (467) | 133,394 (592) | 346,528 (1539) |
| Proportion | 31% | 30% | 38% | 100% |

Source: Authors calculation from study data.

### 5.1.1. Human Costs

The estimated VOSL was XAF 16 Million (USD 71,408), out of which the consumption loss per victim was XAF 7 Million (USD 31,083). The unit human cost was therefore XAF 9 Million (USD 39,964). The total human costs amounted to XAF 133.4 Billion (USD 592 Million) for the year 2018, with slightly more than half of this cost being attributed to serious injuries (see Table 6).

### 5.1.2. Total Production Loss

The loss of production due to death or injuries from RTCs was estimated at XAF 105.1 Billion (see Table 6). Serious injuries account for almost 67% of these costs, with the loss due to fatalities accounting for only 31%. Despite their large numbers, slight injuries are responsible for a negligible loss in production due to the few number of "out of work" days occasioned by slight injuries.

### 5.1.3. Medical Costs

The total medical costs incurred from treating RTCs victims in Cameroon for the year 2018 were estimated to be XAF 107,988 Million (equivalent to US$480 Million, using the PPP of XAF 225 per USD applicable for 2018, see Table 7). The largest proportion of the medical costs is attributable to in-patient treatment (87%) followed distantly by out-patient treatment. Serious injuries account for most (93%) of the medical costs with fatalities and slight injuries accounting for very small proportion (7%). The average treatment duration for a fatality was one (1) day [33], since most fatality cases die after one day of hospitalization. This is the explanation for the negligible share of fatalities in total medical costs. Seriously injured victim spends about 33 days in hospital, and 10 days follow-up with out-patient visits; slightly injured victims spend about four (4) days seeking treatment and mediation on out-patient services, while most fatalities are observed to occur within a day of admission.

**Table 7.** Total medical costs of RTCs in Cameroon in Millions of XAF (USD) for the year 2018.

| Sub-Category | Fatalities | Serious Injuries | Slight Injuries | Total Medical Costs | Proportion by Cost Component |
|---|---|---|---|---|---|
| Ambulance | 48 (0.21) | 604 (2.68) | - | 653 (3) | 1% |
| Emergency services | 20 (0.09) | 254 (1.13) | 688 (3.1) | 963 (4.3) | 1% |
| In-patient treatment | 113 (0.5) | 93,313 (414) | - | 93,426 (415) | 87% |
| Outpatient | - | 6220 (28) | 6726 (30) | 12,946 (57) | 12% |
| Total Medical Costs | 182 (0.81) | 100,391 (446) | 7414 (33) | 107,988 (480) | |
| Proportion by injury level | 0.15% | 93% | 6.8% | | |

Source: Authors calculation from study Data.

### 5.2. Crash-Related Costs

The two components of crash-related costs estimated in this study are vehicle damage and administrative costs.

### 5.2.1. Vehicle Damage Costs

Vehicle damage costs account for the largest proportion (50%) of total RTCs' social costs. The average vehicle damage costs that were adapted and validated in the field are shown in Figure 2. The costs per damaged vehicle in fatal and serious crashes were almost equal. The higher total cost in serious over fatal crashes is explained by the higher number of serious crashes compared to fatal crashes. Although the average damage costs per PDO crash is the lowest (see Figure 2), the PDO crashes account for 50% of the total damage costs (as shown further in Table 8) due to the large number of PDO crashes.

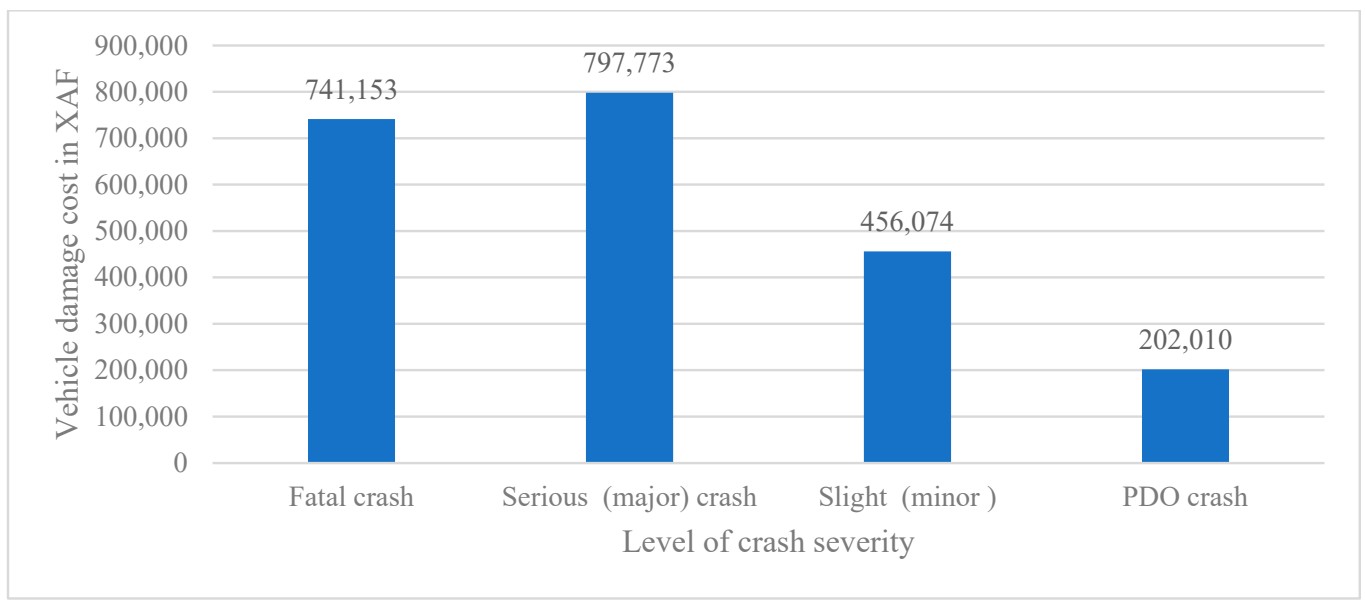

**Figure 2.** Estimated vehicle average damage costs (XAF) per crash by level of crash severity. Source: Authors construction from study data.

**Table 8.** Vehicle Damage Costs (Millions of XAF) in 2018.

| Level of Crash Severity | Vehicle Damage Costs in Millions of XAF (USD Millions) | % |
|---|---|---|
| Fatal crash | 6428 (28) | 2% |
| Serious (major) crash) | 77,373 (344) | 19% |
| Slight | 119,591 (531) | 29% |
| PDO crash | 202,375 (899) | 50% |
| | 405,767 (1802) | 100% |

Source: Authors calculation from study Data.

5.2.2. Administrative Costs (Insurance, Police, Fire Service and Judicial)

The total administrative costs, based on parameters in Table 9, amounted to XAF 56.6 Billion (USD 249 Million) in 2018, comprising only 7% of the total costs (Table 10). Insurance-related expenditure and PDO crashes account for most of the administrative costs: this is based on the large number of vehicle damage crashes and by extension the number of individual compensation cases.

**Table 9.** Unit police and fire service administrative costs by crash severity.

| | Fatal | Serious Injury | Slight Injury | PDO |
|---|---|---|---|---|
| | | Traffic Policemen | | |
| Number at site | 4 | 3 | 2 | 2 |
| Time spent at site (hours) | 0.3 | 0.3 | 0.25 | 0.25 |
| % of crashes attended | 100% | 75% | 10% | 1% |
| | | Firemen | | |
| Number at site | 3 | 3 | 3 | 3 |
| Time spent at site | 0.3 | 0.3 | 0.25 | 0.25 |
| % of crashes attended | 50% | 38% | 1% | 0% |

Source: Authors construction from field survey.

**Table 10.** Administrative costs by sub-category and crash severity in Millions of XAF for 2018.

| Level of Crash Severity/Sub-Category | Administrative Costs in Millions of XAF (USD Millions) | Total |
|---|---|---|
| Total Administrative Costs by Crash Severity | | |
| Fatal crash (1%) | 677 (3) | |
| Serious (major) crash (7%) | 4008 (18) | |
| Slight (8%) | 10,764 (48) | |
| PDO crash (73%) | 41,108 (183) | |
| | | 56,555 |
| Total Administrative Costs by Sub-Category | | |
| Police (1%) | 346 (1.5) | |
| Fire service (0%) | 9 (0.04) | |
| Insurance (97%) | 54,957 (244) | |
| Judicial (2%) | 1243 (5.5) | |
| | | 56,555 |

Source: Authors calculation from study Data.

*5.3. Overall Distribution of RTC Costs by Injury Severity, and Sub-Category of Cost Components*

As stated before, the largest component of the total RTC costs estimated for Cameroon in 2018 is property damage costs (see Table 11) at 50%. This is consistent with patterns observed among LMICs by Wijnen & Stipdonk [4]. The small proportion of total administrative cost corresponds to the pattern in BRS &TRL [18] guide, further emphasizing that they do not form a major part of RTC costs. Human costs are the second largest component accounting for 16% of the costs, which is within the average estimates of LMICs (18%) reported in the analysis by Wijnen & Stipdonk [4]. Production and medical costs each account for 13% of total costs and lower than the averages in the same study.

**Table 11.** Distribution of total social costs of RTCs (XAF) by sub-categories and severity.

| Cost Category | Total Costs in Millions of XAF (USD Millions) | Proportion of Total Cost (Percentage) |
|---|---|---|
| Total Costs Social Cost of RTCs Classified by Sub-Categories | | |
| Medical costs | 107,987 (480) | 13% |
| Production loss costs | 105,147 (467) | 13% |
| Human costs | 133,394 (592) | 16% |
| Property Damage costs | 405,767 (1802) | 50% |
| Administrative costs | 56,556 (251) | 7% |
| Total costs | 808,851 (3600) | 100% |
| Total costs classified by injury severity | | |
| Fatalities | 84,422 (375) | 10% |
| Serious Injuries | 325,155 (1444) | 40% |
| Slight Injuries | 155,791 (692) | 19% |
| PDO | 243,483 (1081) | 30% |
| Total | 808,851 (3600) | 100% |

Source: Authors calculation from study Data.

Slightly more than half (59%) of the total RTCs social costs in Cameroon are attributed to slight and serious injuries. Fatalities account for the least of total costs because of the fewer numbers of fatalities compared to that of injuries. Crashes where no one is injured (PDO level) account for only 30% of total costs in this classification despite the large number of crashes. This could be attributed to the fact that there is no human being who is injured or killed, and therefore no human, medical, production or consumption loss costs are incurred.

### 5.4. Cost Per Casualty

As it is within practice, the sub-category and total RTC costs were expressed on "a per casualty" basis, so as to give an impression on the casualties that largely contribute to the total costs. This is simply achieved by scaling each category of costs with the adjusted number of casualties, all within the same level of injury severity (see Table 12).

**Table 12.** Costs per casualty in XAF (a) and USD (b).

| (a) Category of Costs/Level of Injury | Fatality Crash Costs in XAF (%) | Serious Injury Crash Costs in XAF (%) | Slight Injury Crash Costs in XAF (%) |
|---|---|---|---|
| Medical costs | 35,319 (0%) | 1,560,191 (31%) | 42,618 (5%) |
| Production loss costs | 6,230,875 (38%) | 1,091,628 (22%) | 16,156 (2%) |
| Human costs | 8,743,983 (53%) | 2,088,726 (22%) | 160,671 (10%) |
| Property damage Costs | 1,247,933 (8%) | 1,202,465 (24%) | 687,430 (77%) |
| Administrative costs | 131,510 (1%) | 62,286 (1%) | 61,870 (7%) |
| Total cost per casualty | 16,389,620 | 5,053,288 | 895,515 |
| (b) Category of costs/level of Injury | Fatality crash costs in USD | Serious Injury crash costs in USD | Slight Injury crash costs in USD |
| Medical costs | 157 | 6928 | 189 |
| Production loss costs | 27,688 | 4847 | 72 |
| Human costs | 38,828 | 9275 | 713 |
| Property damage Costs | 5541 | 5340 | 3053 |
| Administrative costs | 584 | 277 | 275 |
| Total cost per casualty | 72,778 | 22,349 | 3977 |

Source: Authors calculation from study data.

From Table 12, the highest cost per casualty is incurred when a person dies in an RTC, with the largest proportion (53%) of this fatality costs attributable to human costs (pain, grief and suffering due to loss of life), followed distantly by loss of production. The negligible fatality medical costs is due to the fact that majority of patients who die from RTCs do so on-scene or within one day hospitalization. Serious injuries are the second largest source of RTC costs, with the largest proportion (31%) of these costs attributed to medical costs. The latter form a significant part of the costs per injured casualty, given the attendant long treatment duration. The costs per slight injury casualty are the least, mostly emanating from vehicle damage (77%).

Lastly, it was not possible to calculate the costs per crash in this study due to the limitations of crash data collection in Cameroon. The crash data records do not differentiate between serious and slight crash, yet the two have different implications on the calibration of the model.

### 5.5. Discussion

The estimated cost of RTCs in Cameroon for 2018 amounts to XAF 808.9 Billion equivalent to 3.8% of GDP, representing a huge leakage from the economy. To put it in perspective, this cost is equivalent to the combined budgetary expenditure on health and infrastructure sectors in 2018. Further, the leakage is equivalent to losing 12% of the total expenditure on all the economic sectors in the same year.

The estimated cost of RTCs exceeds the estimate used in the revised National Road Safety Strategy of Cameroon. The Strategy approximates the cost of road traffic crashes for the period 2013–2017 at XAF 908 Billion, excluding human costs. This implies that the revised strategy assumes an average annual cost of RTCs for this period to be XAF 182 Billion, representing only 22% of the estimated cost. Part of the contributing reasons for the under-estimation is lack of systematic approach in quantifying the costs, and subsequent failure in accounting for massive under-reporting of RTC statistics as demonstrated in this study. The Strategy quotes the estimate of RTC cost to be 1% of GDP based on property damage cost only. However, no attempt is made of adjusting the presumed estimate for

other costs (production, human and administrative) that jointly account for half of RTC costs in Cameroon. Adopting underestimated costs in the revised Strategy undervalues the loss that RTCs impact upon the economy, and may still fail to convince policymakers of the magnitude of the problem. This could lead to inefficient (higher or lower) resource allocation of road safety resources but probably to a repeated cycle of under-investment in road safety in Cameroon as cited by World Bank [2] and quoted in the preceding Strategy.

Property damage cost is the largest (50%) proportion of the total costs. Wijnen & Stipdonk [4] attribute this phenomenon to a lower valuation of human losses in LMICs relative to property valuation so that other components account for larger share. Recent study in Kazakhstan (upper-income economy) by Wijnen [28] estimates human costs at 81% of RTC costs while property damage costs is 11%. It is important to note that the property damage costs in the current study cover only the vehicle, whereas there could be other costs such as loss of property within the vehicle or even loss associated with road-side infrastructure. Lower human costs emanate from the prevailing lower levels of income in Cameroon (LMIC), making income compensation due to a crash negligible. Further, the estimated human costs using GDP-based value transfer do not cover elements such as pain: funeral costs were also not included in the current study further explaining the huge proportion of vehicle damage costs.

Human Costs element is the second largest component at 16% of RTC costs, and is within the range of average for LMICs indicated by Wijnen & Stipdonk [4]. The estimation of this component represents a departure from LMIC literature that leaves out the human costs or estimates the same from insurance payments (see, for instance, Mofandal & Kaniptong [13]). The latter studies use rules of thumb and other methods that do not produce country-specific values for human costs, leading to mostly under-estimation of the human cost element of RTC costs (see Wijnen & Stipdonk [4]).

Medical and production loss are the third and fourth largest proportion of RTCs costs each accounting for 13%. It is apparent that savings amounting up to about XAF 110 Billion can be realized in the health sector, through reduction of RTCs and related injuries. A significant proportion of medical costs is accounted for by serious injuries sustained in RTCs. The savings from reduced RTCs injuries can then be allocated to mitigate other naturally occurring threats to human life, that remain a challenge in Cameroon.

Different data sources (Table A1) considered for the cost calculations had different implications on the overall cost calculations due to the drawbacks associated with the data sources. The police data and hospital data are highly underreported especially for slight and minor injuries, coupled with issues of data loss due to the lack of a centralized accident data registry system which could have led to an underestimation of the total cost of crashes. However, to provide reliable estimates, the police data and hospital were corrected for underreporting. When comparing Police and hospital data sources, the reported injury cases were far greater (more than 96%) for hospital data compared to police data, but the differences in the reported fatality cases were not too significant with the police data recording more cases (6% higher). We then used Police data source for estimating fatalities and hospital data source for estimating injuries. Similar accuracy levels for injuries and fatalities from police and hospital based data sources were observed also in other African countries [30].

Other sources of crash data like insurance companies were not considered due to higher unreliability, but rather this data was useful for the administrative costs. Moreover, insurance data was not segregated according to different crash types and could not be used as a reliable source of crash data. Other data sources could not provide reliable primary data, for example, the hospital for the medical cost. As such, this study relied on other published data from similar countries (Guinea and Sudan) to provide near estimates which could either underestimate or overestimate the costs.

There was a generally low response rate in the planned surveys and an inability to collect field data due to socio-economic instabilities in some areas during the study period. Travel restrictions occasioned by domestic and international lockdowns linked to

the COVID-19 pandemic also prevented the researchers from making a physical follow-up of data in the field. This shortcoming was addressed by considering experts opinions.

## 6. Conclusions and Recommendations

This study sought to estimate the socio-economic cost of RTCs in Cameroon for the year 2018 using a combination of approaches (human capital, willingness to pay and restitution costs), following international guidelines. The study makes an attempt to correct for under-reporting of RTCs, providing evidence of the RTC cost in a Sub-Saharan LMIC setting.

The estimates of this study indicate that the socio-economic burden of RTCs in Cameroon is equal to 3.8% of GDP in 2018, higher than that obtained using near similar approach in LMICs setting. However, proportions of cost components in total RTCs cost match those in LMICs. The estimate is also higher than presumed costs used in road safety policy such as the prevailing National Road Safety Strategy of Cameroon. The study indicates that under-reporting of RTCs data in Cameroon has serious implications on the apparent value of RTCs costs on the economy and needs urgent redress. The study recommends the following:

1. Adoption of systematic methods in estimation of RTCs for purposes of road safety policy, as this will generate a better perspective of the consequences of RTCs.
2. Maintaining accurate data by instituting changes in handling of RTCs administrative data: priority should be given to operationalizing the recently established centralized RTCs database and outsourcing RTCs data processing in the police and Gendarmerie to hasten data processing. Accurate data is essential for proper assessment of socio-economic costs as well as tracking progress in Road Safety commitments such as that of SDG target 3.5.
3. Primary surveys are recommended in repeat studies to overcome the lack of data that was resolved by using transfer values in the current study.
4. Similar studies within Sub-Saharan African context are recommended to provide data for purposes of analysis in continental road safety policies.

Overall, the models developed in this study for the costs estimation are suitable for application in countries of similar context to Cameroon, and caution should be made when replicating the models to other contexts. Future research should aim at factoring the effects of missing components such as funeral costs, loss in employee productivity upon return to work and prison costs for those road users that go to prison due to causing a road crash, and in addition, it would be beneficial to independently to calculate the social costs using solely WTP or HC approach and comparing the estimates with those provided in this study. Surveys that will permit the use of WTP need to be carried out following international best practices.

**Author Contributions:** Conceptualization, P.T.; methodology, P.T. and W.W.; validation, D.S.U. and M.M.K.; formal analysis, P.T. and M.M.K.; investigation, N.B.K. and P.T.; resources, P.T.; data curation, P.T. and N.B.K., M.M.K.; writing—original draft preparation, P.T.; writing—review and editing, D.S.U., P.T., M.M.K. and W.W.; supervision, L.P.; project administration, P.T. All authors have read and agreed to the published version of the manuscript.

**Funding:** This research received no external funding; it was self-funded by some of the authors.

**Informed Consent Statement:** Informed consent was obtained from all subjects involved in the study.

**Data Availability Statement:** The data that support the findings of this study are available from author P.T., upon reasonable request.

**Conflicts of Interest:** The authors declare no conflict of interest.

## Appendix A

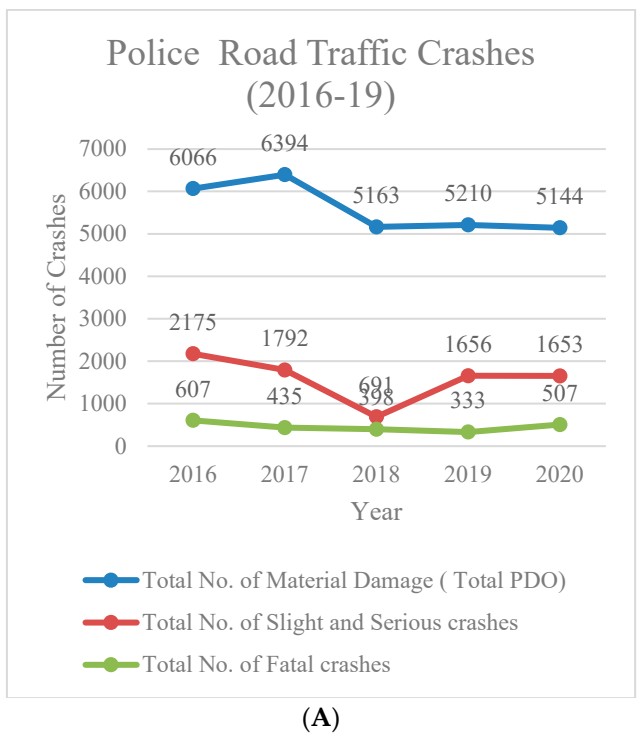 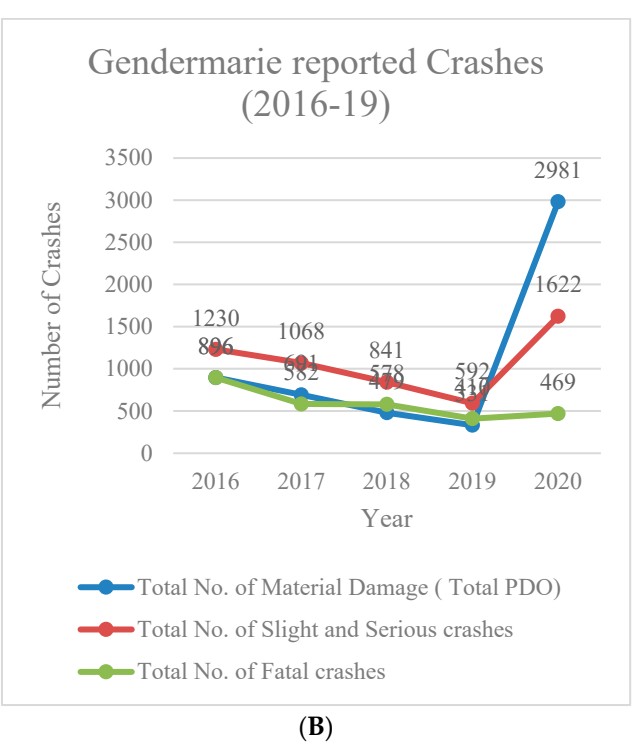

(**A**)                                                          (**B**)

**Figure A1.** RTCs reported by the Police (**A**) and Gendarmerie (**B**): 2016–19. Source: Authors Construction from Agency data (Police and Gendarmerie) Cameroon.

**Table A1.** List of data needed to calculate each cost item and data sources.

| Designation | Nature and Description of Severity | Data Source |
|---|---|---|
| Number of reported road casualties | Fatalities, by gender age categories of fatalities | Obtained from the National Health Observatory Previous hospital survey [29] provided information on gender and age structure of RTC Injuries |
| | Serious injuries, by gender age categories of serious injuries | Obtained from the National Health Observatory Previous hospital survey [29] provided information on gender and age structure of RTC Injuries |
| | Permanently disabled | Obtained from the National Health Observatory Previous hospital survey [29] provided information on gender and age structure of RTC Injuries Applied ratios from Mofandal & Kaniptong (2016) on the injuries to obtain and estimate of permanently disabled. |
| | Slight injuries, by gender and age categories | Obtained from the Ministry of Public Health's National Health Observatory in Cameroon Previous hospital survey [29] provided information on gender and age structure of RTC injuries |

**Table A1.** *Cont.*

| Designation | Nature and Description of Severity | Data Source |
|---|---|---|
| Number of reported crashes | Fatal crashes<br>Serious injury crashes<br>Slight injury crashes<br>Property damage only crashes | Gendarmerie National, Police Reports<br>Used the ratios obtained for injuries to allocate injuries between the two |
| Number of casualties per crash | Fatal crashes: number of fatalities, serious injuries and slight injuries per fatal crash<br>Serious injury crashes: number of serious injuries and slight injuries per serious injury crash<br>Slight injury crashes: number of slight injuries per slight injury crash | Calculated from Gendarmerie and police reports |
| Percentage of underreporting of the number of casualties/crashes | Fatalities<br>Serious injuries<br>Slight injuries<br>Fatal crashes<br>Serious injury crashes<br>Slight injury crashes<br>Property damage only crashes | WHO estimation of fatalities in Cameroon<br>Injuries to crash ratios from police data<br>Household survey information |
| Medical—Transportation | Number of ambulance trips for road casualties<br><br>Costs of an ambulance trip | Proportion obtained from hospital survey on casualties by<br>Household survey,<br>Adapted from Contingent Valuation Study by Lee et al. [38] DRC, PPP conversions used |
| Medical—Hospitalization | Average number of days of hospital treatment for hospitalized injuries<br><br>Average cost of emergency treatment per patient<br><br>Average costs of in-patient hospital treatment (with overnight stay) per patient per day | Adapted from Hospital Survey in Ethiopia [17] Household survey, Casualty survey<br>Adapted from Mofadal&Kaniptong (2016), converted to current prices and XAF<br>Adapted from Mofadal & Kaniptong (2016), converted to current prices and XAF |
| Medical—Out-patient treatment | Number of slight injuries that have been treated on the hospital emergency department (without overnight stay)<br><br>Average costs of hospital emergency treatment (without overnight stay) | Proportion of slight injuries in derived from McGreevy et al. (2014)<br><br>Derived from Mofadal & Kaniptong [13], converted to 2018 prices and PPP exchange rates |
| Medical—Inability to work | Average number of days a seriously injured casualty is unable to work after the crash<br>Number of days a slightly injured casualty is unable to work after the crash | Adapted from Murad et al. (2011) and complemented with Casualty survey<br>Adapted from Murad et al. (2011) and complemented with Casualty survey |
| Production and consumption loss | Population by<br>gender<br>age (preferably each age: number of 0, 1, 2, 3–99 years)<br>education level<br>Life expectancy by gender and age<br>Age of entering the labour market by gender and education level<br>Average retirement age by gender<br>Gross Domestic Product per capita<br>Yearly number of working hours per labourer, by gender<br>Official discount rate used in economic assessments of governments investments (for example used in cost-benefit analyses) | National Institute of Statistics<br><br>National Institute of Statistics<br>National Institute of Statistics<br><br>National Institute of Statistics<br>National Institute of Statistics<br>National Institute of Statistics<br><br>Ministry of Finance |

**Table A1.** *Cont.*

| Designation | Nature and Description of Severity | Data Source |
|---|---|---|
| Human costs | GDP per capita<br>Private consumption per capita,<br>VOSL ratios | National Institute of Statistics<br>National Institute of Statistics<br>Milligan et al. [22] |
| Vehicle damage | Average damage per vehicle in fatal crash<br>Average damage per vehicle in serious injury crash<br>Average damage per vehicle in slight injury crash<br>Average damage per vehicle in PDO crash | Adapted from Ethiopia [17] for vehicles and Pakistan for motorcycles [35]. Data updated using relevant price indices and PPP exchange rates |
| Insurance administration costs | Administrative costs of insurance companies related to vehicle insurances (personnel costs, overhead) | Association of Insurance data &Sudan [13] |
| Police costs | Proportion of crashes (by severity) the police attends<br>Average number of policemen coming to the place of a road crash, by crash severity<br>Average time a police officer spends per crash, by crash severity<br>Average wage of a police officer | Police interview, complemented by household surveys<br>Police interviews<br>Police interviews<br>Police interviews |
| Fire emergency services | Proportion of crashes attended by fire services department<br>Average number of personnel coming to the place of a road crash, by crash severity<br>Average time spent per crash, by crash severity<br>Average wage of fire officers | Police interviews; fire service interviews |
| Legal | Time spending police on prosecution of offenders who caused a crash<br>Number of trials concerning road users who have caused a crash<br>Average costs per trial<br>Number of road users that go to prison due to causing a road crash<br>Average number of days in prison<br>Prison costs per prisoner per day | Police interviews<br>Interviews judicial organization (e.g., law courts)<br>Interviews judicial organization (e.g., law courts)<br>Interviews judicial organization (e.g., law courts, prisons)<br>Interviews judicial organization (e.g., law courts, prisons)<br>Interviews judicial organization (e.g., law courts, prisons) |

Source: Authors own construction from study data.

**Table A2.** Characteristics of Countries used for transfer values.

| | Cameroon | Sudan | Ethiopia | Guinea | Congo |
|---|---|---|---|---|---|
| *Road safety and motorization indicators* | | | | | |
| age limit on imported vehicles in years | 7 [1] | 5 | none | 8 | 7 |
| motorization/1000 | 3235 | 3165 | 692 | 2095 | 2483 |
| estimated fatalities by WHO/100,000 | 30.1 | 25.7 | 26.7 | 28.2 | 27.4 |
| health coverage index | 44 | 43 | 39 | 35 | 38 |
| expenditure on health% GDP | 5% | 6% | 4% | 5% | 5% |
| *Social Economic Properties* | | | | | |
| Class (WB) * | LM | LM | L | L | L |
| UN-HDR ** | L | L | L | L | L |
| UN-Statistics *** | D | D | D | D | D |

Source: Author compilation from WHO, World Bank and UN databases. * H = high, UM = upper middle, LM = lower middle, L = low income country. ** L: Low human development. *** D: Developing region. [1] Even if the official age limit for car importation in Cameroon is seven years, the fleet age is very old as explained in the context of this study, making the comparison with Ethiopia valid.

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
