# Peer review of "An Assessment of the Social Costs of Road Traffic Crashes in Cameroon"

_sustainability, doi:10.3390/su15021316_

Round 1

Reviewer 1 Report

This study assesses the social cost of road crashes in Cameroon. The topic is important and there are some new findings.

My main concern is the use of human capital approach without adjusting for pain, suffering, grieve, inconvenience, etc. These are real costs that sufferers incur. Would you pay more than the medical cost to avoid someone breaking your legs? These issues need to be discussed in both the methodology section as well as discussion section.

There are several sweeping statements and loose logic throughout the paper that needs it be tighten or qualified. Please check through the paper carefully, especially the introduction, discussion and conclusion.

For example, "The quantified costs can therefore inform trade-off in allocation of national resources to competing needs and interests, leading to allocation of adequate resource to road safety strategies". Being informed does not necessarily lead to allocation of adequate resources, especially if there is a budget constraint. It may lead to more informed choices but not necessarily to adequate allocation. Being uninformed may also lead to over-allocation in contrast to your claim. There are several such statements made throughout the paper.

Another example and I will stop giving them. "This approach of planning for road safety without evidence is likely to worsen an already fragile position of road safety in Cameroon. For instance, while Africa’s fatality per 100,000 population attributable to RTCs increased by 10 % between 2010 and 2013 (from 24.1 to 26.6), that of Europe 54 decreased by 10 % (from 10.3 to 9.3) during the same period (WHO, 2015)." There are many factors contributing to crashes, one part of the claim does not support the other part of the claim. 

There are also many implicit assumptions that are used but not stated and discussed. These assumptions relate to using specific ratios and methods for estimation. You are implicitly assuming that they are accurate for Cameroon, which is debatable. Besides these assumptions there is also an assumption that the use of average is appropriate. The are strengths and weaknesses of using average versus median. For example, bias due to outliers. Did you check the distribution in Cameroon? Is average wage a good proxy for the cohort of casualties in Cameroon? Are crashes income, education or occupation related? Please state the various assumptions clearly and provide some discussion on each.

What is a Gendarmere? Is it something similar to rural police like RCMP in Canada?

There are too many tables and some can be combined. The most important one are not even shown or incomplete. I would like to see a final table or figure showing the total cost per casualty and per crash for different severity levels (also in US dollar).

Why is there little or no medical cost for fatality (0%)? How is fatality defined in this study? Are most casualties dead at the crash site?

Pleas clean up all the typos and errors. For example, "Error! Reference source not found" should not occur in a professionally prepared manuscript.

Reviewer 2 Report

Article title:  An assessment of the Social costs of Road Traffic Crashes in Cameroon

This research seems very interesting, but I have some comments about improving this paper.

Abstract

Problem statement should be clearly highlighted.

Please add the main findings and recommendations of the study.

Introduction

Add the literature section separately based on the findings of the studies from developed and developing countries. The literature section should clearly show the research gap your study wants to fill.

Use the journal's recommended reference style, i.e., [1], [2], etc.

Methodology

Figure 1: Elaborate in the text. Write the complete form of RTCs in figure 1

Equation 1 – 11: Add a brief explanation and purpose of using each equation clearly.

Data

Table 1: Add the agency name in the source rather than just writing “agency”—same comments for other tables if required.

Line 325: Error! Reference source not found…. Check and correct it.

Results and Discussion

Check the section number and correct them because section 3 is about “Data,” and authors also written section 3 with results and discussion. Moreover, the authors added a separate section as discussion. It would be better to add a discussion part in “result and discussion” section that confronts developed and developing countries' realities.

Results need to be explained more with specific reasons and arguments.

Conclusions and recommendations

Recommendations should be clearly highlighted.

Limitations of the study should add in the conclusion part.

Future research should be added.

Reviewer 3 Report

Interesting article, which is an attempt to estimate the costs of road accidents in a situation of limited data.

Remarks:
Line 25 Please explain LMICs abbreviation on first use.
Line 31 Please explain RTIs abbreviation on first use.
Row 156 Medical, not the Medial.
Formula 3, line 170 variable names mismatch.
Line 187 Why was the decrease in employee productivity upon return to work omitted?
Line 211 - variable name in the formula inconsistent with the description in line 227
Line 212 - please explain with GDP / Cap - if it is gross domestic product per capita, please enter notation without the slash sign
Line 213 - Unexplained variables
Line 214, line 215 - editorial flaws - bold, parentheses
Lines 240, 247, 255 - please explain where the data on the proportions used in the formulas were obtained (PAsev, FAsev, Prop)
Line 320 redundant space.
Line 325 - Reference error.
Lines 372 - 381 - please explain how the repair valuation was accepted - authorized service station or other workshop?
Line 439 - please do not hyphenate the words in the header, the word outpatient should be capitalized like the other words.
Line 449 unnecessary space.
Line 453 - useless space before the parenthesis.
Line 464 - in my opinion the involvement of emergency services is significantly underestimated. In my opinion, the travel time is often greater than the times presented. Please analyze the travel time, rescue operation time, documentation of the incident on site, preparation of documentation at the police station, preparation of firefighting equipment for the next operation or conducting detours in the event of a long road closure. In my country, there are at least 6 firefighters in accidents with injured people. In the event of a fatal accident, there are definitely more policemen on the spot, and the inventory of the accident scene sometimes takes several hours. This part requires review and improvement.
Line 491, 492 - please correct quotation marks
Appendix A - the table shows the costs of the perpetrators of the incidents in prison, and it is not included in the costs of accidents. Why?
Appendix A - out-patient is spelled as one word elsewhere.
Table A14 - please explain the term "age limit in years"
Table A14 - Please do not split country names in the header.

There are many editorial errors in the article. Please correct them while preparing the revised version of the manuscript.

The citations do not follow the MDPI style. Please correct!

Good luck!

Round 2

Reviewer 1 Report

The authors have improved the manuscript but there are still several typos and errors that need to be cleaned up.

Also, there are still several sweeping statements and loose logic throughout the paper that needs it be tighten or qualified. Please check through the paper carefully, especially the introduction, discussion and conclusion.

For example, “The reliability of these ratios was confirmed by comparing them” – confirmed is too strong. First, your estimates are only consistent with theirs and relative comparison cannot confirm that you are right. Second, you are also assuming that their estimates are correct. I suggest that you tone down such claims (throughout the paper). I suggest something like “Our estimates are consistent with …., which provides (some) support that they are (maybe) reliable.

Another example is “The study demonstrates that under-reporting” – again, demonstrates is too strong. It implies that you have direct evidence from your data and analysis or concrete evidence but the evidence provided are indirect and circumstantial although very indicative.

There are still several key assumptions besides the using the average that need to be clearly stated and discussed. I suggest checking each step of the calculation. Discussion should include why the assumption is made or why this the method chosen, as well as the possible sources of bias and how it would affect the estimates (under-estimate/over-estimate).

For example, the main source of data on fatality under-reporting for the study is the WHO. You are assuming that WHO estimates are correct. If I remember correctly, the estimates are obtained from a Poison regression using a cross sectional data from different countries and the accuracy is not that high, which is expected as it does not include country-specific data but uses only very generic across the board information. The confidence intervals are therefore quite large. The actual estimated value thus may be much higher or much lower.

A similar example is the assumption that the hospital survey in Guinea is a good estimate for the under-reporting rate in Cameroon. Please justify and discusses its implications on your estimates.

Is pain, grieve and suffering limited to production loss? We all suffer from these in any injuries, regardless of whether we are contributing to production or not. Are you assuming that children and the elderly do not suffer from pain and grieve? Do we not grieve for our loved ones who are injured or killed? Do we not take leave from work because our loved ones are injured or killed? How about inconvenience, time cost and other intangible costs? If my car is damage and under repair, don't I have to suffer from these non-out-of-pocket costs? This study deals mainly with explicit costs and ignores implicit and intangible costs. This assumption should be clearly stated at the beginning of the methods sections. The implications on the estimates should also be discussed in the results or discussion section.

Assumptions about data also need to be stated and discussed. Besides police and hospital data, there are other sources such as insurance data. What are the relative strengths and weaknesses of the different sources and what are the implications on the estimates (over or under) for the data source you chosen?

Reviewer 2 Report

The authors incorporated the commments.

Author Response

We thank the reviewer for reviewing our manuscript.